# Climate suitability predictions for the cultivation of macadamia (*Macadamia integrifolia*) in Malawi using climate change scenarios

Emmanuel Junior Zuza[1]*, Kadmiel Maseyk[1], Shonil A. Bhagwat[2], Kauê de Sousa[3,4], Andrew Emmott[5], William Rawes[5], Yoseph Negusse Araya[1]

1 Faculty of Science, Technology, Engineering & Mathematics, School of Environment, Earth and Ecosystem Sciences, The Open University, Milton Keynes, The United Kingdom, 2 Faculty of Arts & Social Sciences, School of Social Sciences and Global Studies, The Open University, Milton Keynes, The United Kingdom, 3 Faculty of Applied Ecology, Agricultural Sciences and Biotechnology Inland Norway University of Applied Sciences, Department of Agricultural Sciences, Hamar, Norway, 4 Digital Inclusion Area, Biodiversity International, Maccarese, Italy, 5 The Neno Macadamia Trust, Bedford, The United Kingdom

* Emmanuel.Zuza@open.ac.uk

**Data Availability Statement:** The data and code underlying this study are available on Zenodo

## Abstract

Climate change is altering suitable areas of crop species worldwide, with cascading effects on people reliant upon those crop species as food sources and for income generation. Macadamia is one of Malawi's most important and profitable crop species; however, climate change threatens its production. Thus, this study's objective is to quantitatively examine the potential impacts of climate change on the climate suitability for macadamia in Malawi. We utilized an ensemble model approach to predict the current and future (2050s) suitability of macadamia under two Representative Concentration Pathways (RCPs). We achieved a good model fit in determining suitability classes for macadamia (AUC = 0.9). The climatic variables that strongly influence macadamia's climatic suitability in Malawi are suggested to be the precipitation of the driest month (29.1%) and isothermality (17.3%). Under current climatic conditions, 57% (53,925 km$^2$) of Malawi is climatically suitable for macadamia. Future projections suggest that climate change will decrease the suitable areas for macadamia by 18% (17,015 km$^2$) and 21.6% (20,414 km$^2$) based on RCP 4.5 and RCP 8.5, respectively, with the distribution of suitability shifting northwards in the 2050s. The southern and central regions of the country will suffer the greatest losses ($\geq$ 8%), while the northern region will be the least impacted (4%). We conclude that our study provides critical evidence that climate change will reduce the suitable areas for macadamia production in Malawi, depending on climate drivers. Therefore area-specific adaptation strategies are required to build resilience among producers.

(https://zenodo.org/record/5249199#.YSj_
m45KhEY).

**Funding:** The research was funded by The Open University and The UK Research and Innovation through Global Challenges Research Fund (GCRF). I would like to highlight that the funders had no role in this study.

**Competing interests:** The authors have declared that no competing interests exist.

## 1. Introduction

Ecosystems, human health, livelihoods, food security, water supply, and economic growth are all impacted by global climate change [1]. The severity of these effects is predicted to increase in direct proportion to the degree of global warming. By the 2050s, it is estimated that a 2°C increase in warming will increase the number of people exposed to climate-related risks and poverty by several hundred million [1]. This warming presents significant threats to many parts of Africa's current agricultural production systems, particularly among smallholder farming families with limited adaptive potential [2,3]. Sub-Saharan Africa (SSA) is one of the most vulnerable regions to climate change due to decreased amount and distribution of precipitation and increased temperatures [4–6]. Malawi is particularly vulnerable to climate change because of its high poverty level, limited cash flow and technological infrastructure [7]. Moreover, the country is heavily reliant on the rain-fed agricultural sector for food security and economic development [8].

Agriculture is the backbone of Malawi's economy and society [9]. Malawi's growing food demand, on the other hand, will make it more difficult to meet in the coming decades, as already stressed agricultural systems are threatened by population growth and rising incomes [10]. Therefore, knowledge of how climate change may alter crop production patterns and their climate suitability (hereinafter "suitability") is critical for effective agricultural adaptation in Malawi. Multiple studies in the country have already indicated the dire consequences of climate change on crop production. For example, Bunn et al. [11] and Dougill et al. [6] have predicted losses in suitable areas for tea production in the low-lying areas of the Thyolo district. Climate change is expected to reduce maize yields by at least 50% [8,12,13] and tobacco yields by at least 45% [14]. Tobacco is the mainstay of the rural economy in Malawi, contributing to almost 40% of the country's exports earnings [15]. Given the current downturn in tobacco market trends, macadamia has been identified as a suitable tobacco alternative that may contribute more to Malawi's economy [22]. Nonetheless, this will be achievable only if suitable areas for macadamia cultivation are identified and mapped under current and future climate conditions.

Macadamia is a perennial crop native to Australia [16]. As a result, the crop is vulnerable to climate influences such as sudden temperature shifts and variations in precipitation which diverge away from current and historic growing conditions found in its native habitat. Economic macadamia production is therefore only possible within certain geographical and climatic ranges [17]. Optimum diurnal and seasonal temperatures for macadamia are within the ranges (14 °C by night and 30 °C by day), with prolonged periods outside this range having adverse effects on growth, yield, and quality [18–20]. Regarding precipitation, macadamia grows healthy and is productive in areas with well-distributed rainfall, totaling an average of 1500mm per year [21]. Water stress during nut maturity has negative impacts on the yield and quality of macadamia [22]. To stimulate flowering and nut set, macadamias require strong temperature contrasts and mild water stress for up to four months [22,23]. This demonstrates how much macadamia production is influenced by climate, while geographical parameters such as altitude, aspect, and slope are only considered important in terms of affecting temperature and water requirements [24].

Understanding macadamia's current and future suitability is essential for developing mitigation and adaptation strategies for the projected negative impacts of climate change, especially among smallholder producers (those with less than one hectare of land) in Malawi. For these smallholders, the promotion of macadamia agroforestry remains a viable adaptation option. This is because the farmers may intercrop their macadamia trees with annuals, enhancing their long-term resilience to climate change. Evidence suggests that climate change is

already reducing macadamia suitable areas [25], limiting yields (quality and quantity) [21,26,27], and increasing pests and diseases globally [28]. Though it is assumed that climate change is likely to reduce suitable areas for macadamia [17,24], integrated spatially quantitative impact studies are still lacking.

This study aims to fill this gap. We present evidence of the impact of climate change on the suitability of macadamia in Malawi. We applied an ensemble modelling approach driven by 17 General Circulation Models (GCMs) under two emission scenarios (RCP4.5 and RCP8.5) for the 2050s. We were particularly interested in examining the potential distribution of macadamia areas in Malawi, identifying the key determinants of macadamia, and measuring the crop's response to climate change. Such climate risk assessments on the macadamia sector are essential for generating scientific evidence on the impacts of climate change, particularly among smallholders with little adaptive capacity. In addition to informing policy and trade, this assessment is a first step toward identifying and implementing adaptation measures tailored to macadamia within global boundaries. We concentrate on climate projections for the 2050s to align with the United Nations framework of global challenges in agriculture and food security [29].

## 2. Methodology

### 2.1. Study area

We examined the suitability for macadamia in Malawi, a southern African country that falls within the longitudes 30 and 40 and the latitudes −17 and −10. The country spans over ~118, 484 km$^2$, with 94, 449 km$^2$ (80%) of land area and 24, 035 km$^2$ (20%) of water surfaces. The country is divided into three main regions; Central, Southern and Northern parts, with 28 districts (S1 Fig, S2 Table) with varying elevations. Because of variations in topography (Fig 1), parental materials (soil), and management, soil nutritional status varies greatly across the country, particularly among smallholder farmers [30].

Malawi has a subtropical climate with two distinct seasons: the rainy season (November to April), which accounts for 90–95 percent of the annual precipitation, and the pronounced dry season (May to October) [15]. The rainy season varies by region; for example, rains begin earlier in the southern region than in the central region, and the north has less pronounced dry seasons, especially at higher elevations. Furthermore, the geographical distribution of temperature and precipitation in Malawi is determined by its topography and proximity to the Indian Ocean and Lake Malawi. Average annual precipitation ranges from 500mm in low-lying marginal areas to over 3000mm in high plateau areas [31]. Malawi's mean annual minimum and maximum temperatures are 12 and 32˚C, respectively, with the lowest temperatures in June and July and the highest in October or early November [32].

Fig 2 illustrates the spatial pattern of average annual temperatures (a) and annual precipitation (b).

### 2.2.Occurrence data

Data on macadamia tree species' occurrence was collected from smallholder macadamia farms in Malawi during the 2019/20 growing season through a field survey. For our analysis, we only sampled ten-year-old successfully established macadamia orchards under smallholder rainfed conditions. We focus on ten-year-old macadamia orchards because the productivity of macadamia depends on the age of the orchard (i.e., the yield of the crop increases with age) [25], and at this age, the crop is at the start of peak production. A total of 120 orchards were sampled throughout Malawi, but only 84 locations were used for this study. This is because we resampled the occurrence points to a tolerance of 5 km so that no two points could be found in one

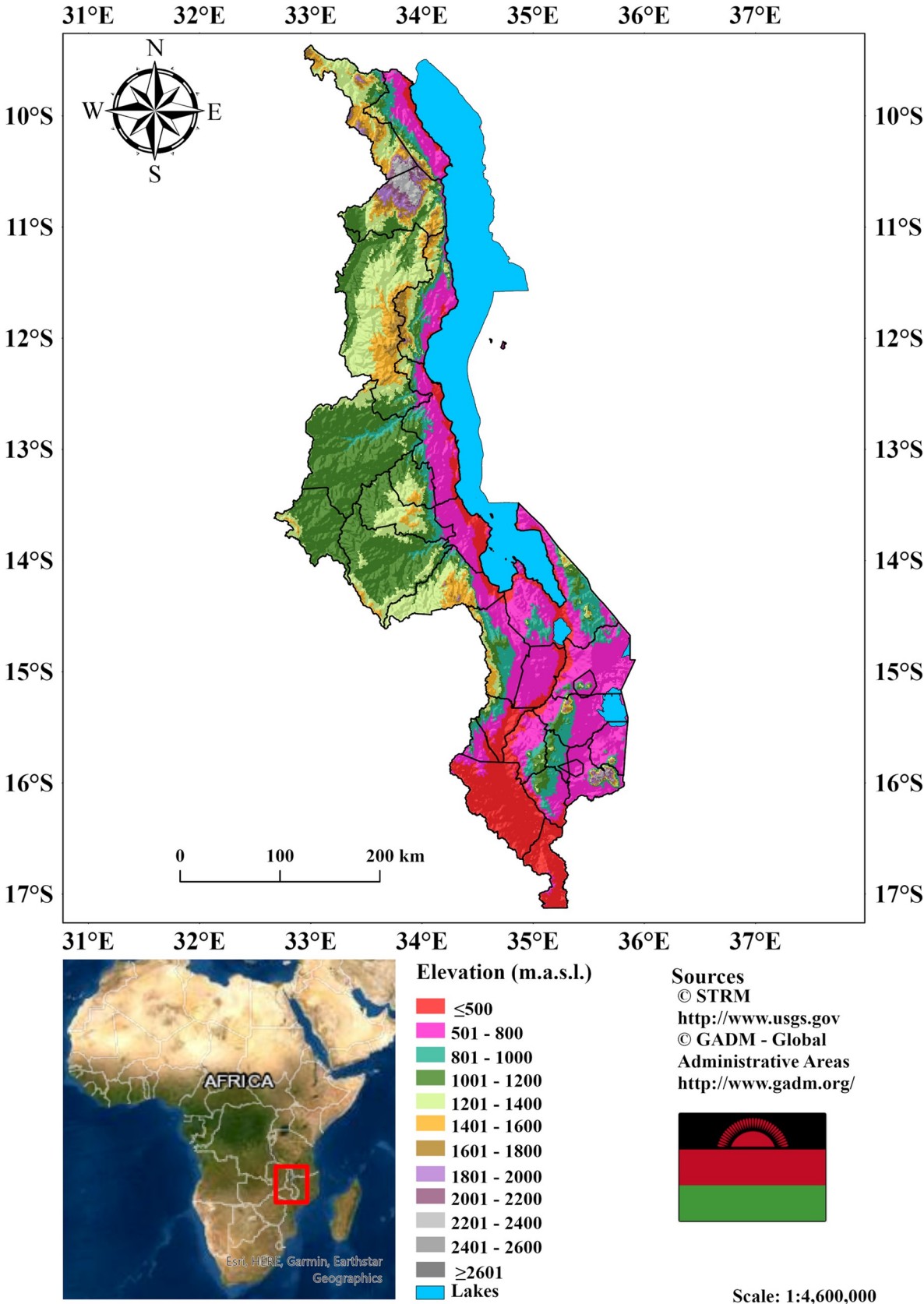

**Fig 1. Geographic location and topography of Malawi based on Shuttle Radar Topography Mission digital elevation model data.**

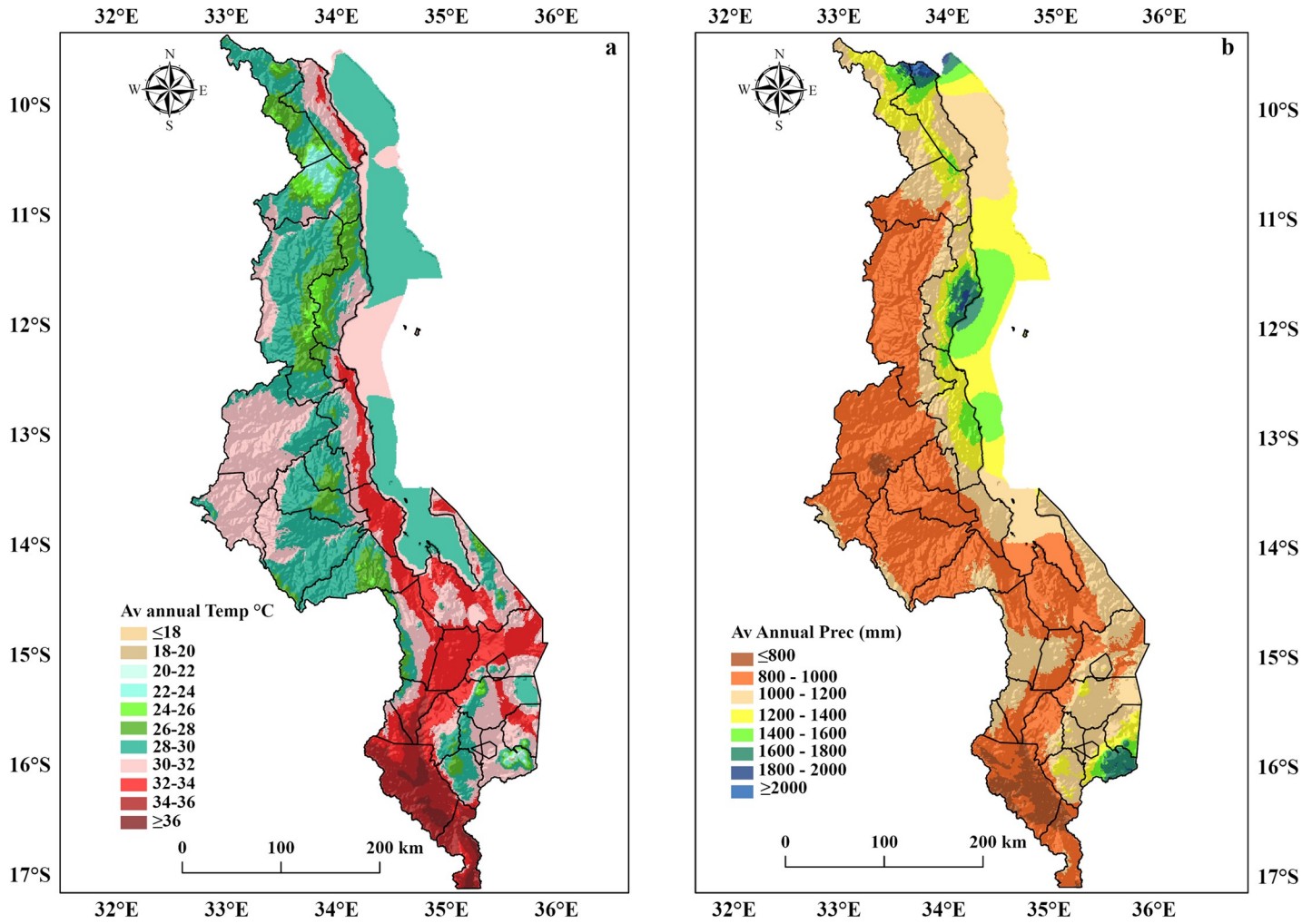

**Fig 2.** a) Average annual temperature ($^{\circ}$C) and b) Precipitation (mm) of Malawi based on WorldClim-global climate data.

environmental layer at a resolution of 5 km x 5 km. At each farm, the Global Position System (GPS) coordinates (in WGS84 datum) were collected using a global position system (Garmin eTrex Vista® Cx) together with altitude. Additionally, utilizing the approach described by Barbet-Massin et al. [33], we generated background pseudo-absence points (Fig 3) to cover any sampling biases in the study.

### 2.3. Climate data

We used bioclimatic predictors (~1970–2000) from WorldClim data set version 1.4 (http://www.worldclim.org/) at a spatial resolution of ~ 5 km x 5 km to model the current areas suitable for macadamia in Malawi (the data was clipped to the Malawian country boundary). Calculated from monthly temperature and precipitation climatologies, these bioclimatic variables describe spatial variations in annual means, seasonality, and extreme/limiting conditions (S3 Table). We utilized bioclimatic variables derived from 17 GCMs (to reduce the uncertainty inherent within individual GCMs) (S4 Table) based on two RCPs (S5 Table) of climate change for our future predictions [34]. We selected RCP 4.5, which is an intermediate scenario that considers an intermediate greenhouse gas (GHG) concentration and predicts an average

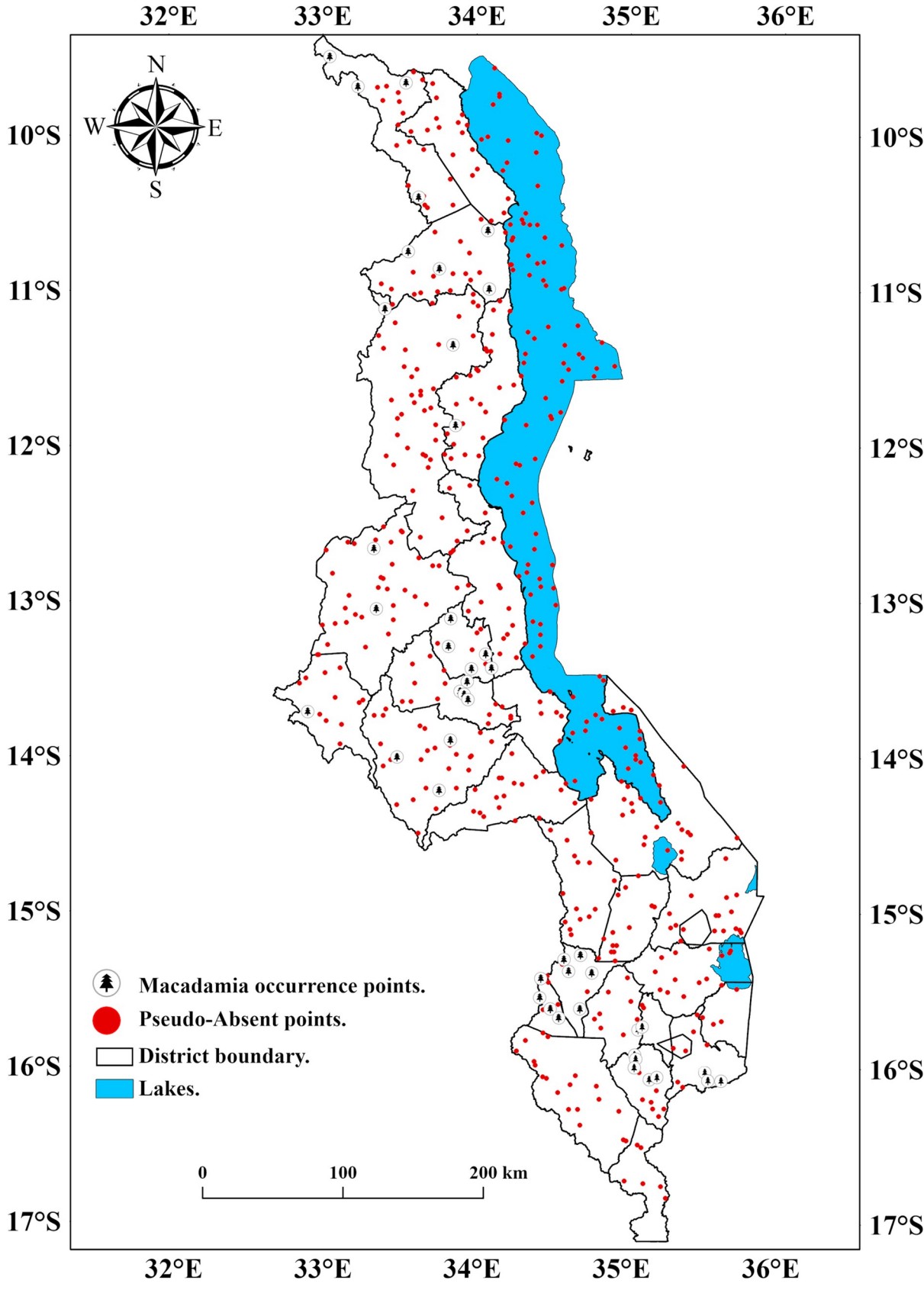

**Fig 3. Map of Malawi showing macadamia occurrence points and pseudo absent points.**

increase in temperature by 1.4 °C (0.9–2.0 °C) and RCP 8.5, the most pessimistic scenario, which considers higher GHG emissions concentration with a 1.4–2.6 °C projected increase in mean global temperature by the 2050s (period 2040–2060).

For this study, we did not consider scenario 2.6 because it represents the most efficient and effective mitigation scenario, i.e., keeping the temperature below 2 °C. At present, this scenario is not feasible with projections of current policies (expected temperature increase of 3.3–3.9 °C) [35]. Furthermore, to achieve this scenario, emissions would need to be 25% lower than in 2018 (GHG emissions rose by 1.5% in 2018) [36]. Emission scenario 6.0 was also not considered for our analysis because its projections are between the ranges of RCP 4.5 and RCP 8.5 [37]. Further, RCP 6.0 only has 42% of the GCM outputs, meaning that the scenario has fewer outputs than the rest of the emission scenarios [38].

## 2.4. Variable selection

In species distribution models, multicollinearity (multiple correlations between variables) among the bioclimatic predictors may result in overfitting or bias in the resulting suitability model [39,40]. To avoid these challenges, variable quality evaluation criterion using a multicollinearity degree was employed through the variance inflation factor analysis (VIF). The VIF indicates the degree to which the standard errors have been inflated due to the levels of multicollinearity among the independent variables used in running the model [41]. VIF is directly calculated from a linear regression model with the focal numeric variable as a response, as shown in Eq (1).

$$VIF = \frac{1}{1 - R_i^2} \qquad (1)$$

Where $R^2$ is the regression coefficient of determination of the linear model.

In our study, the "*ensemble.test*" function inherent in the "*BiodiversityR*" package available in R [42] was used to eliminate correlated variables. Following the recommendation made by Ranjitkar et al. [43], we retained variables that had a VIF of less than 10 (Table 1).

## 2.5. Modelling approach

We modelled macadamia's current and future distribution in Malawi based on an ensemble suitability method implemented by the R package "*BiodiversityR*" [44]. We used an ensemble modelling technique because it combines predictions from various algorithms and can provide better accuracy in predictions than relying on individual species distribution models [45]. The procedure consisted of four steps.

**Table 1. Bioclimatic variables used in the final suitability model and their variance inflation factor (VIF).**

| Variable name | Bioclimatic variable | Unit | VIF Score |
|---|---|---|---|
| Bio 14 | Precipitation of driest month | mm | 2.96 |
| Bio 3 | Isothermality (Bio2/Bio7) x 100 | - | 1.51 |
| Bio 15 | Precipitation seasonality (cv x 100) | - | 3.25 |
| Bio 2 | Mean diurnal range | °C | 6.05 |
| Bio 18 | Precipitation of warmest quarter | mm | 5.23 |
| Bio 13 | Precipitation of wettest month | mm | 2.12 |
| Bio 6 | Minimum temperature of the coldest month | °C | 2.02 |
| Bio 4 | Temperature seasonality (Standard deviation x 100) | - | 1.61 |

We evaluated the predictive accuracy of 18 algorithms of species distribution models (SDM) using a cross-validation technique in the first stage. The SDM algorithms used in our analysis were those that can distinguish between suitable and non-suitable areas without needing absence locations [35]. Following work by Brotons et al. [46] and Thuiller et al. [47], we divided the occurrence data into two distinct sets by randomly assigning 70% of the data as a training dataset to fit the model, and the remaining 30% were used as test data to evaluate the model's predictive accuracy. A five-fold (partition) cross-validation replicate was performed in each of the model algorithms to evaluate the stability of the prediction accuracy as described by Rabara et al. [48] and Mudereri et al. [39]. Each SDM algorithm's performance was evaluated from each partition separately after individual algorithms were assessed with data from the other four partitions. Cross-validation validates the performance of models and prevents overfitting, particularly in cases where the amount of data may be limited [39,49].

The area under the curve (AUC) criterion computed by the R package "*PresenceAbsence*" [50] was used to evaluate the performance of each algorithm. The AUC value is a specific measure of model performance that demonstrates the model's ability to locate a randomly chosen present observation in a cell with a higher probability than a randomly selected absence observation [45,48]. Based on the recommendation by Kindt and Cole [42], we used an AUC value of 0.77 as a threshold to select the best-performing algorithms for this analysis. SDM algorithms that did not meet this criterion were not used to calculate the final ensemble model's suitability [51]. AUC values of 0.75 are considered reliable, 0.80 as good, and 0.9 to 1 as having excellent discriminating ability [52].

We utilized the presence-only approach for our study, and this is because, for agricultural applications of niche models, it is inappropriate to treat areas without current production as entirely unsuitable. Further, determining whether a species is absent in a specific location is difficult and rare, so absence data may not be a true representation of naturally occurring phenomena [53]. As an alternative, we randomly generated 500 background pseudo-absence points for our analysis. A caveat to this approach is the recommendations of Barbe-Massin et al. [33] regarding the use of lower pseudoabsences in some algorithms. Then, we combined these background pseudo-absence points with the 84 occurrence points "presence only" for the niche modelling of macadamia.

The second step consisted of retaining only the algorithms that contributed at least 5% to the ensemble suitability ($S_e$) [43]. This procedure generated AUC values for each and the parameters of the response functions (model training) to estimate the probability values of species occurrence based on the climate of each grid cell of the study area. The AUC values for the selected SDM algorithms are shown in Table 2. The results of all the models were then combined by calculating for each the weighted average (weighted by AUC for each model) of the probability values from each model to generate the ensemble suitability map. The AUC values obtained by each algorithm were weighted using the following equation:

$$Ensemble\ (S_e) = \frac{\sum_i W_i S_i}{\sum_i W_i} \tag{2}$$

Where the ensemble suitability ($S_e$) is obtained as a weighted (*w*) average of suitabilities predicted by the contributing algorithm ($S_i$).

The predicted suitable area for the probability of macadamia was calculated using threshold values, i.e., $\geq 0.34$ for the suitable area, while $< 0.34$ was regarded as unsuitable [39]. To generate the probability maps, we used the maximum sensitivity (true positive[+]) and maximum specificity (true negative[-]) approach [54], where we reclassified the probability maps to a binary raster image (suitable/unsuitable areas). Then, using the Malawi shapefile in R, the

**Table 2. Performance evaluation of the ensemble model.**

| Algorithm | Method | AUC |
|---|---|---|
| Envelope model | BIOCLIM | 0.86 |
| Multivariate distance | DOMAIN | 0.90 |
| Additive models: Generalized additive models | GAM | 0.89 |
| Regression: Multivariate adaptive regression splines | MARS | 0.93 |
| Stepwise GAM | GAMSTEP | 0.85 |
| Mahalanobis distance | MAHAL | 0.99 |
| Maximum entropy | MAXENT | - |
| Boosted regression models: Generalized boosted regression models | GBM | - |
| Generalized linear models | GLM | 0.98 |
| Support vector machines | SVM | 0.86 |
| Stepwise boosted regression tree models | GBMSTEP | 0.94 |
| Artificial neural networks | NNET | 0.96 |
| Random Forest | RF | 0.94 |
| Multivariate Adaptive Regression Splines | EARTH | - |
| Stepwise generalized linear models | GLMSTEP | 0.82 |
| Mixed GAM Computation Vehicle | MGCV | 0.85 |
| Support vector machines | SVM | 0.98 |
| Flexible discriminant analysis | FDA | - |
| Ensemble | ENSEMBLE | 0.90 |

predicted binary values for each pixel were extracted. Finally, the total number of pixels for each predicted class was used to estimate the total coverage of the predicted suitable area against the unsuitable area within Malawi. Following recommendations by Chemura et al. [55], we divided the two suitability classes (suitable/unsuitable) into five classes (unsuitable, marginal, moderate, optimal, and highly suitable). The final visualization maps for the suitability classes of macadamia were developed using Arc GIS Pro software version 2.5 (https://arcgis.pro/).

In the fourth stage, we applied the derived baseline suitability model to each of the 17 downscaled GCMs to predict the future distribution of suitable areas for macadamia by the 2050s. Finally, the results of the 17 GCMs probability layers were integrated into a single layer, using the criterion of likelihood scale [56,57], which requires at least 66% of agreement among GCMs to keep the predicted presence or absence in a given grid cell. The final visualization maps for the future suitability classes of macadamia were developed using Arc GIS Pro software version 2.5 (https://arcgis.pro/).

## 3. Results

### 3.1. Model performance evaluation

Our results show that the ensemble model performance (AUC = 0.9) was sufficient for our modelling activity when measured using the AUC. The model's evaluation revealed that the modelling of macadamia areas in Malawi was based on model competence rather than chance (Table 2). Importantly, the high AUC value provides confidence to apply the ensemble model for examining the areas suitable for macadamia under current and future climatic conditions.

### 3.2. Contribution of variables to the suitability of macadamia

The importance of climatic factors driving the suitability of macadamia production in Malawi is shown in Fig 4. Precipitation-related variables are the most important in determining

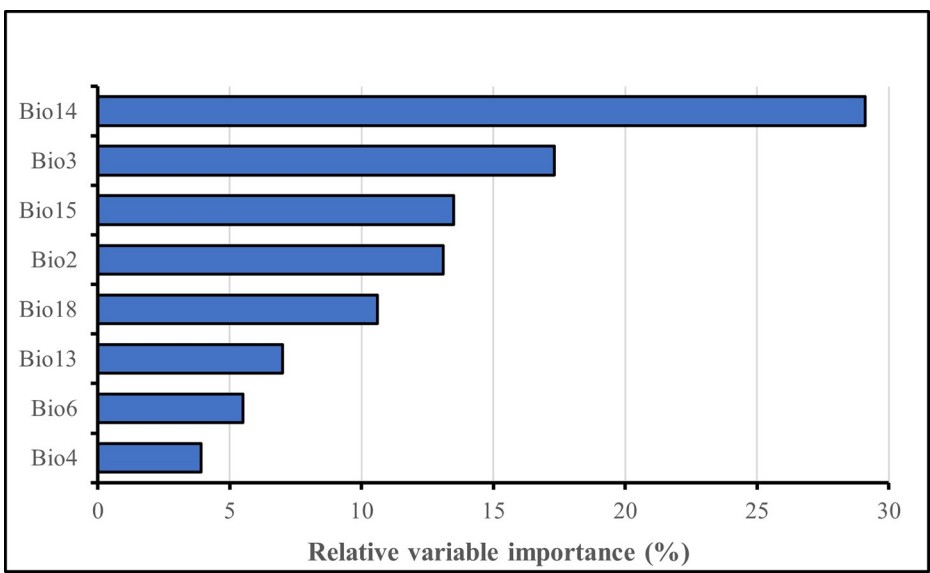

**Fig 4. The importance of a variable in explaining macadamia suitability in Malawi.** Data is obtained from the averages of the 18 species distribution model algorithms.

suitability for macadamia in Malawi and contributed 60.2% towards macadamia suitability. Precipitation of the driest month (May–November) and precipitation seasonality accounted for more than 40% in determining the suitability for macadamia. Precipitation of the driest month is the variable with the greatest relative influence (29.1%) on the suitability for macadamia. Temperature variables contribute 39.8% towards macadamia suitability in Malawi. Among the temperature variables, isothermality (17.3%) (calculated by dividing mean diurnal temperature range by mean annual temperature range) was the most significant. Our model results found that annual means do not affect the suitability for macadamia production in Malawi.

### 3.3. Current suitability for macadamia in Malawi

Results of the present (~1970–2000) suitability analysis reveal that 57% (53,925 km$^2$) of the surface area in Malawi is suitable for macadamia production, with the largest area (25.8%, which is 24,327 km$^2$) in the central region of the country (Table 3, Fig 5). Of the 57% that is suitable, optimal suitability (26%, 24,565 km$^2$) is observed in the highland parts of the country with elevations ranging from 1000–1400 m.a.s.l. Notably, in some parts of Dowa, Chitipa, Mulanje, Mwanza, Mzimba, Ntchisi, Nkhatabay, Rumphi, and Thyolo districts (S2 Table). Moderate suitability (22.4%, 21195 km$^2$) is projected in the mid-hills between 950–1000 m.a.s.l. in the districts of Blantyre, Chiradzulu, Dedza, Kasungu, Lilongwe, Mchinji, and Zomba. Marginally suitable areas were found to be in the lower elevated ($\leq$ 900 m.a.s.l) parts of Malawi. Because of the topography, the districts of Neno and Ntcheu have both optimal and marginally suitable areas for macadamia (Fig 5). Furthermore, according to our model

**Table 3. Area and percentage suitable for growing macadamia under current climatic conditions.**

| Region | Area (km$^2$) | Percentage (%) |
|---|---|---|
| Central | 24,327 | 25.8 |
| Northern | 19,341 | 20.5 |
| Southern | 10,257 | 10.7 |
| Total | 53,925 | 57 |

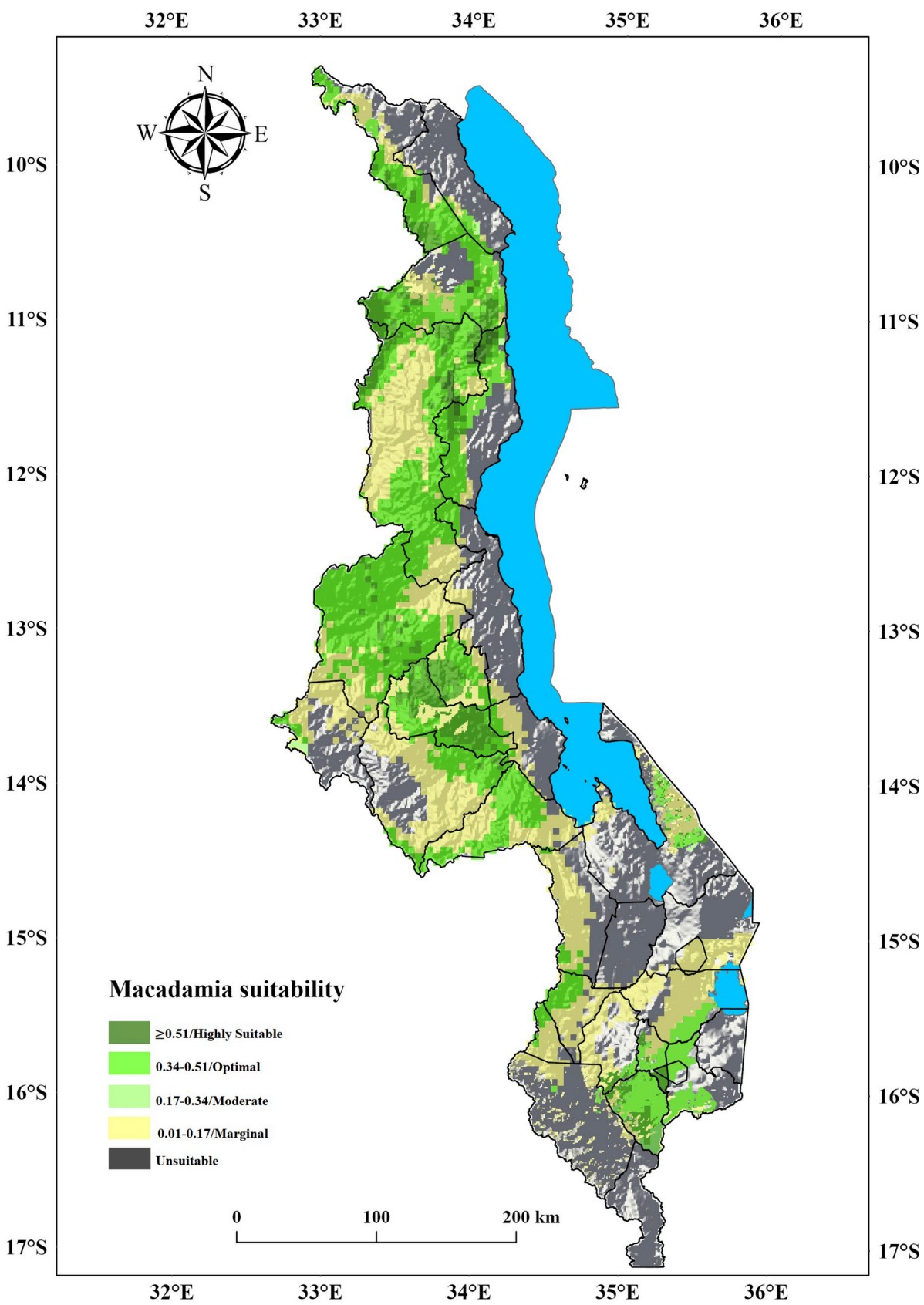

**Fig 5. Current suitability for macadamia production in Malawi.** The model results were exported into Arc GIS Pro Software version 2.5 to generate the map in this figure.

projections, the existing distribution of climatically suitable areas for macadamia closely matches the crop's occurrence areas.

### 3.4. The impacts of climate change on macadamia in Malawi

The impacts of climate change on macadamia suitability in Malawi are depicted in Table 4, Fig 6. By the 2050s, the extent of suitable areas for macadamia is projected to decrease under both emission scenarios utilized in this study. Our projections show a net loss of 18% and 21.6% (Table 4) of suitable areas for macadamia production under RCP 4.5 and RCP 8.5, respectively. This translates to 17,015 km$^2$ (RCP 4.5) and 20,414 km$^2$ (RCP 8.5) of Malawi's total cultivatable surface area. Lower altitude areas (0–900 m.a.s.l.) will experience the greatest decline in suitability. These losses will be more pronounced in Malawi's southern region, estimated to lose between 81.7% (RCP 4.5) and 85.2% (RCP 8.5) of all its current suitable areas due to projected drier and hotter conditions in the next coming decades. Due to climate change, the Thyolo district, which is currently Malawi's most productive and largest macadamia growing area, is expected to lose 100% (1228 km$^2$) of its suitable areas for macadamia production. In addition, the ensemble model predicts that the area suitable for macadamia in the country's central region will shrink by at least 7.2% (6,784.1 km$^2$) (RCP 4.5) and 8.4% (7,950.1 km$^2$) (RCP 8.5). For the northern region of Malawi, the suitability for macadamia is predicted to decline by 2% (1,850 km$^2$) and 4% (3,730 km$^2$) under RCP 4.5 and RCP 8.5, respectively.

Despite the projected losses in suitable areas for macadamia production due to climate change, our predictions suggest that 39.1% (36,910 km$^2$) and 35.5% (33,511 km$^2$) of Malawi's surface area will remain suitable for the crop under RCP 4.5 and RCP 8.5, respectively (S6 Table). The results from the intermediate scenario show that 18.6% (17,543 km$^2$), 18.5% (17,491 km$^2$), and 2% (1,876 km$^2$) of Malawi's cultivatable areas will remain suitable for macadamia production in the 2050s in the central, northern, and southern regions, respectively (Fig 7, S7 Table). The outcomes for the pessimistic scenario suggest that approximately 17.3% (16,377 km$^2$), 16.5% (15,611 km$^2$), and 1.6% (1,523 km$^2$) of Malawi's land will remain suitable for macadamia in the central, northern, and southern regions, respectively. In addition, based on RCP 4.5 and RCP 8.5, our model predicts an average gain in suitable areas of +0.22% (207 km$^2$) and +0.5% (476 km$^2$). These newer areas are expected to occur in Dedza (Mua and Chipansi), Mangochi (Namwera and Chaponda), Salima (Kasamwala), and Thyolo (Thekerani) districts. However, these only apply to a small portion of the country and cannot compensate for the country's decreased suitability.

## 4. Discussion

### 4.1. Contribution of variables to the suitability of macadamia

Precipitation and temperature have been identified as critical factors influencing crop growth and yields worldwide [53]. We find that in Malawi, suitability for macadamia is influenced by

**Table 4. Simulated impacts of climate change on macadamia suitability in Malawi.**

| Region | RCP 4.5 | | RCP 8.5 | |
|---|---|---|---|---|
| | Area (km$^2$) | Percentage (%) | Area (km$^2$) | Percentage (%) |
| Central | 6,784.1 | 7.2 | 7,950.1 | 8.4 |
| Northern | 1,850 | 2.0 | 3,730 | 3.9 |
| Southern | 8,380.9 | 8.9 | 8,733.9 | 9.2 |
| Total | 17,015 | 18 | 20,414 | 21.6 |

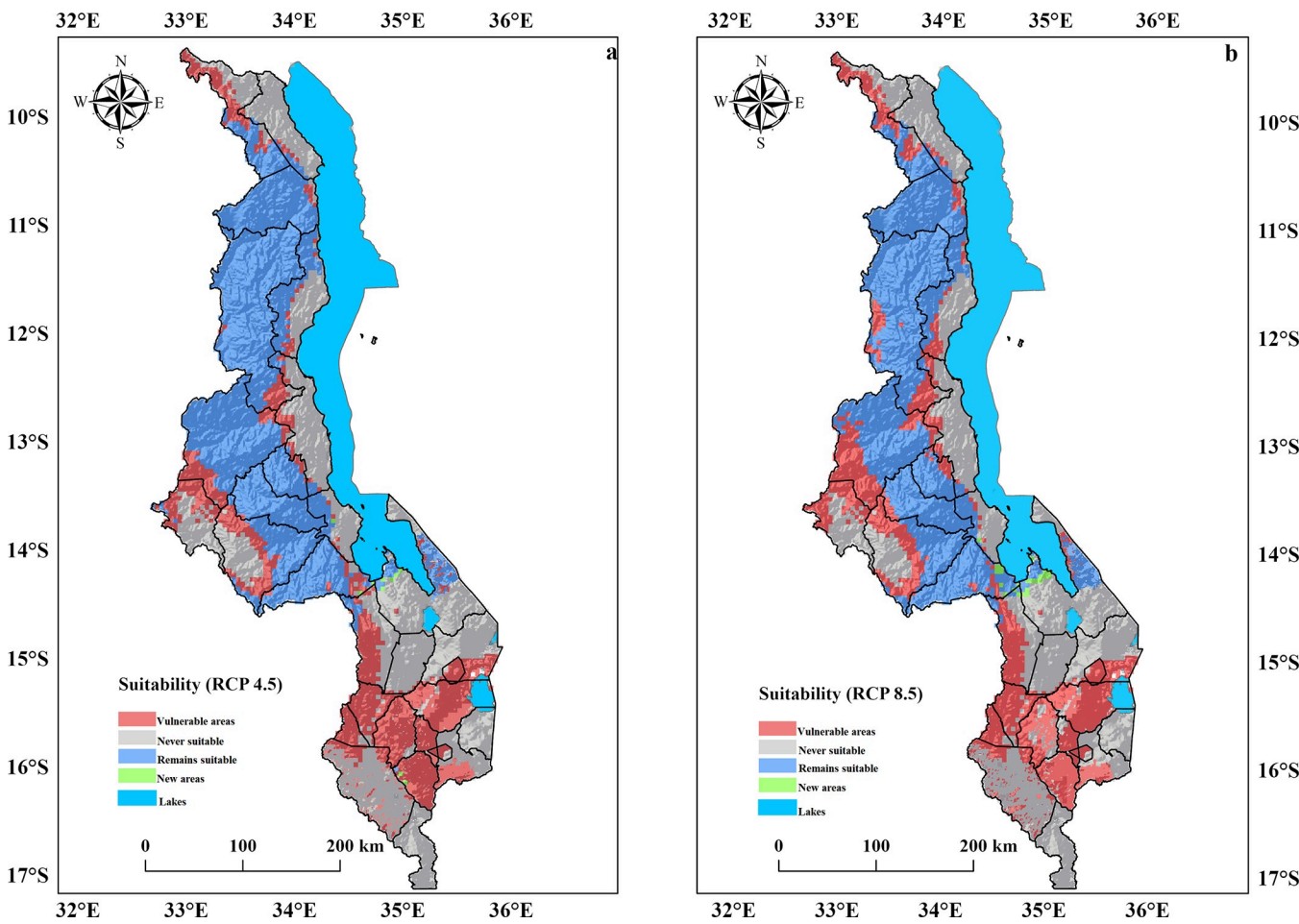

**Fig 6.** Shifts in macadamia suitability due to climate change by 2050 (a) RCP 4.5 (b) RCP 8.5. The model results were exported into Arc GIS Pro Software Version 2.5 to generate the map in this figure.

precipitation, temperature, and seasonal variations of these two factors rather than the annual means, confirming a previous report by Evans [58]. However, climatic variables identified in our study differ from climate indicators for macadamia on a global scale [58]. Conversely, in Nepal, temperature-based factors were identified primarily as determinants for the suitability for macadamia [25]. Chemura et al. [40] argue that differences in scale and geography explain such variations, implying that local and regional factors can influence macadamia potential. This explains our findings, which show that precipitation-based parameters are more relevant in predicting macadamia suitability than temperature-based factors, verifying zoning studies for macadamia production done for the country [59].

According to our results, the precipitation of the driest month (May–November) and precipitation seasonality are the two most essential precipitation variables that affect the suitability of macadamia in Malawi, according to our results (Fig 4). Our results reveal that the dry season in Malawi concurrently coincides with the flowering, nut development, and oil accumulation stages in macadamia growth. Moisture stress, on the other hand, is detrimental to macadamia growth and development. Mayer et al. [60] found that moisture stress inhibits and delays flower development in macadamia, thereby reducing the nut yields and quality. Moreover, water stress induces premature nut drop in macadamia, which affects the yields negatively

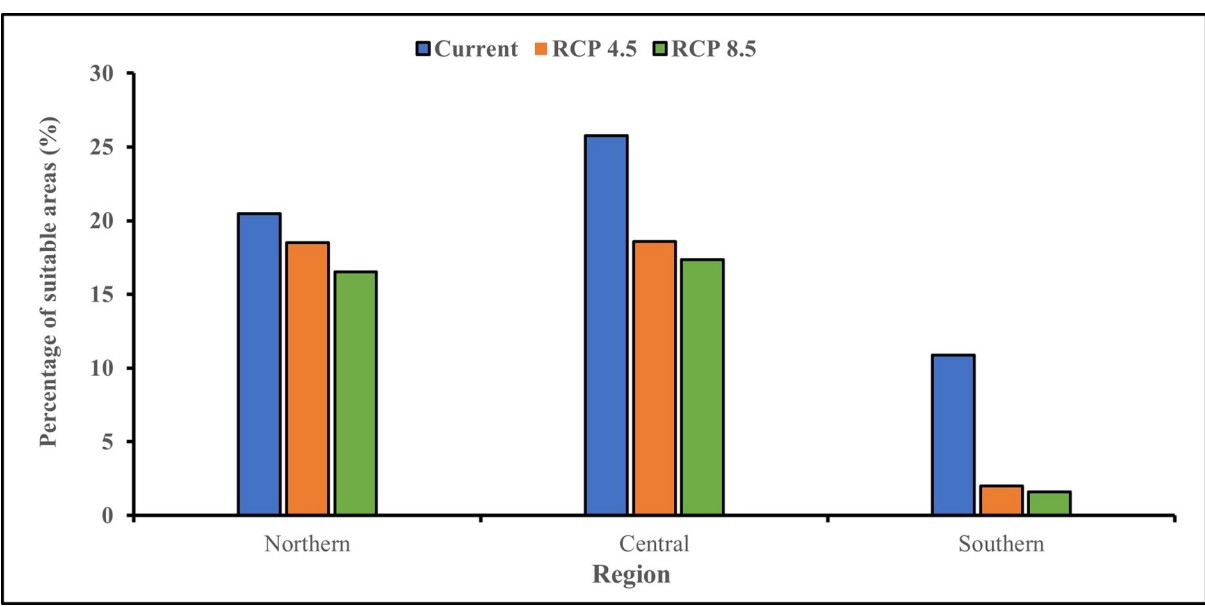

**Fig 7. Percentage of predicted suitable areas for macadamia production using current and future climate scenarios.**

[22]. In Australia, Nagao et al. [58] found that water deficits from prolonged drought periods caused macadamia flower loss and tree mortality. Consequently, projections that climate change will decrease the number of rainy days and months [61,62], thus reducing moisture availability to the macadamia trees during the dry season in many parts of Malawi will drive many areas out of macadamia production. These findings confirm and, more importantly, extends the work by Dougill et al. [6], who predicted that climate change will decrease the amount and distribution of precipitation throughout Malawi, particularly the southern region, altering the suitability of important perennial crops such as tea, coffee, and macadamia in the country. Farmers are therefore encouraged to adopt moisture conservation measures (mulching, rainwater harvesting, box ridging, and basins) and possibly develop irrigation infrastructure to meet the water requirements for macadamia growth, particularly during the drier months of the year.

Isothermality (17.3%) and the mean diurnal range (13.1%) are two other important factors influencing macadamia suitability in Malawi (13.1 percent). Our findings suggest that large fluctuations in day and night temperatures, as well as increased warming ($\geq 30$ $^{o}$C), are responsible for the marginal suitability for macadamia in Malawi, notably along the lakeshore and Shire valley, confirming previous research [25,27,63,64]. Such temperature increases result in increases in evapotranspiration, which raises the crop water requirements of macadamia, especially during critical phenological stages. Higher day temperatures of more than 30 $^{o}$C have already been linked to excessive water loss from the macadamia plants [58]. Such moisture losses result in a disproportional supply of nutrients within the macadamia nut, limiting oil buildup and negatively affecting the nut quality [21]. As a result, predictions that climate change will increase the number of days (30.5 days per year) with temperatures above 30 $^{o}$C and hot nights (40 days per year) with temperatures above 14 $^{o}$C [65], will undoubtedly reduce the number of suitable areas for macadamia production in Malawi. Subsequently, irrigation will be crucial for long-term macadamia production, especially during the hotter, drier months (May-November), to compensate for water lost through evapotranspiration.

## 4.2. Impact of climate change on macadamia suitability in Malawi

The results of our analysis reveal that extensive areas in Malawi under the current climatic conditions are suitable for macadamia production (Table 3, Fig 5). Moreover, our outcomes suggest potentially suitable growing areas for macadamia in Malawi's south-eastern parts outside the current producing zones. The suitability maps depict possible production areas, some of which have not yet been translated to realized areas [53]. This also suggests the broad adaptability of some macadamia cultivars that allow their production from high potential areas to marginal and low input areas with several environmental constraints. Nonetheless, because of their limited buffering capacity, these areas are the most vulnerable to climate change.

Malawian regions are already falling outside the recommended optimal range (14–30 °C) for macadamia production, which is attributed to the increase in annual mean temperatures (0.9 °C) and overall drying recorded in the past five decades [56,57]. According to our analysis, climate change is likely to reduce the suitable areas for macadamia production in the 2050s in Malawi (Table 4, Fig 6). The lowlands, predominantly those in the southern region, will be the most vulnerable to these losses ($\geq$ 85%), with suitability shifting towards the country's central and northern regions. The decreases in suitable areas are attributed to the projected increases in the intensity and frequency of heatwaves, droughts, and temperatures linked to the El Niño Southern Oscillation [66]. Barrueto et al. [25] predicted losses in suitable areas for macadamia production in Nepal's lowlands due to warming conditions caused by climate change, concurrent with the current study results for Malawi. In Ethiopia, Chemura et al. [40] projected declines in suitable areas for specialty coffee under climate change scenarios, confirming our results that climate change may have a negative impact on crop suitability. Bunn et al. [11] predicted losses in suitable areas for tea production in southern Malawi due to projected increases in warming and frequency of droughts, which is consistent with the current study in the same region.

Our study shows that suitable areas for macadamia production in the northern region will face minor losses ($\leq$4%). This is because a larger percentage of the region (75%) is located at higher elevations (Fig 1), making it less vulnerable to temperature changes than the country's central and southern regions. Further, we observe losses in suitability in some high elevated (1400 m.a.s.l.) areas in the northern and central regions. The decrease is due to projected increases in cloud cover [7], resulting in less light reaching the trees, thereby reducing total net photosynthesis for tree growth and oil accumulation, subsequently affecting nut yields and quality. In addition, heavy cloud cover has been reported to cause thick shells (making shelling difficult and expensive) in macadamia and lowers the overall nut yields and quality [17].

Our findings, therefore, show the sensitivity of macadamia to variations in environmental conditions. Farmers can thus continue planting macadamia trees in areas where no changes in suitability for macadamia are expected. However, both research and field-based evidence from discussions with farmers show that climate-related changes are already occurring and affecting the suitability for macadamia production in Malawi. Farmers are, therefore, encouraged to start implementing adaptation measures such as the use of improved macadamia varieties, agroforestry, intercropping, water conservation, and irrigation for long-term and sustainable macadamia production. Nevertheless, these suitability changes are predicted to occur over the next 30 years, so these will mostly impact the next generation of macadamia farmers. Therefore, there is still time for adaptation. Failure to adapt in time to the risk of decreasing yields and incomes may lead to migration, food insecurity, and reduced incomes among the producers.

## 4.3. Applicability and potential limitations of this study

Species distribution modelling is founded on assumptions intrinsic in the models, some of which cannot be tested [67,68]. Although this study's findings can be considered robust,

several issues should be considered in interpreting and applying the results. Though we identified areas as suitable for macadamia production based on environmental predictors, however on the ground, this may not directly translate to the size of the arable land. In addition, other physical (soil physical and chemical factors) and socio-economic factors (including the gender and age of the smallholder farmers, availability of agricultural advisory services, access to roads, and market availability) which are used in determining the suitability of an area for crop production were not considered in our analysis. It is therefore recommended to take extra caution when using the results of this study. Nonetheless, the results of this analysis are important for future planning purposes. Therefore, there is a need for a thorough evaluation of adaptation approaches suggested for smallholder macadamia farmers, as these may be different from those utilized by commercial growers.

## 5. Conclusions

An ensemble model was used in this study to determine Malawi's current and future suitability for macadamia production. The study's findings lead to three important conclusions. For starters, precipitation is the most important determinant of macadamia suitability in Malawi. Second, the current and future macadamia production areas identified exist on agricultural land currently used to grow other crops. As a result, we propose promoting macadamia intercrops and agroforestry as a climate change adaptation strategy. Third, the extent of suitable areas for macadamia production in Malawi is projected to decrease under both emission scenarios utilized in this analysis, and the most vulnerable areas are those in southern Malawi. Thus, we conclude that the macadamia sector faces production risks from climate change, but there are opportunities for adaptation strategies to build a resilient sector in Malawi.

## Supporting information

**S1 Fig. Map showing the districts of Malawi.**
(TIF)

**S1 Table. Suitable climatic conditions for macadamia production in Malawi.**
(DOCX)

**S2 Table. Regions and districts in Malawi.**
(DOCX)

**S3 Table. Bioclimatic variables available in WorldClim.**
(DOCX)

**S4 Table. The general circulation models (GCMs) used to obtain climatic variables under scenarios RCP 4.5 and RCP 8.5 in 2050.**
(DOCX)

**S5 Table. Characteristics of climate change scenarios by the 2050s (RCPs).**
(DOCX)

**S6 Table. Future distribution area of macadamia production in Malawi by 2050.**
(DOCX)

**S7 Table. Areas that will remain suitable for macadamia production by the 2050s by region.**
(DOCX)

**S8 Table. Summary of news reports about climate change affecting macadamia production worldwide (period 2013–2019).**
(DOCX)

## Acknowledgments

Thanks, should also go to Prof. Rick Brandenburg, North Carolina State University, USA, Dr. Michael G. Chipeta, Oxford University, Dr. Edith B. Milanzi, MRC Clinical Trials, University College London, the Neno Macadamia Trust, the U.K. and Highlands Macadamia Cooperative Union Limited (HIMACUL) smallholder farmers, Malawi. We further express our gratitude to Mr. Ken Mkangala and Nicholas Evans for their constructive comments and feedback on the state of macadamia production in Malawi. However, mistakes and omissions are our responsibility.

## Author Contributions

**Conceptualization:** Emmanuel Junior Zuza, Andrew Emmott, Yoseph Negusse Araya.

**Data curation:** Emmanuel Junior Zuza, Kauê de Sousa.

**Formal analysis:** Emmanuel Junior Zuza, Kauê de Sousa.

**Funding acquisition:** Kadmiel Maseyk, Shonil A. Bhagwat, Andrew Emmott, Yoseph Negusse Araya.

**Methodology:** Emmanuel Junior Zuza, Kadmiel Maseyk, Shonil A. Bhagwat, Kauê de Sousa, Andrew Emmott.

**Project administration:** Kadmiel Maseyk, Yoseph Negusse Araya.

**Resources:** Shonil A. Bhagwat, William Rawes.

**Software:** Emmanuel Junior Zuza, Kauê de Sousa, William Rawes.

**Supervision:** Kadmiel Maseyk, Shonil A. Bhagwat, Andrew Emmott, William Rawes, Yoseph Negusse Araya.

**Validation:** Emmanuel Junior Zuza, Kadmiel Maseyk, Shonil A. Bhagwat, Kauê de Sousa, Andrew Emmott, Yoseph Negusse Araya.

**Visualization:** Emmanuel Junior Zuza, Kadmiel Maseyk, Shonil A. Bhagwat, Andrew Emmott, William Rawes, Yoseph Negusse Araya.

**Writing – original draft:** Emmanuel Junior Zuza, Andrew Emmott.

**Writing – review & editing:** Kadmiel Maseyk, Shonil A. Bhagwat, Kauê de Sousa, Andrew Emmott, William Rawes, Yoseph Negusse Araya.

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
