## [Decision Letter · Decision Letter 0]

22 Jun 2021

PONE-D-21-15846

Climatic suitability predictions for the cultivation of macadamia in Malawi using climate change scenarios.

PLOS ONE

Dear Dr. Zuza,

Thank you for submitting your manuscript to PLOS ONE. After careful consideration, we feel that it has merit but does not fully meet PLOS ONE’s publication criteria as it currently stands. Therefore, we invite you to submit a revised version of the manuscript that addresses the points raised during the review process.

Our reviewers noted that the manuscript is interesting and well presented but will require some revisions. Key issues raised by the reviewers is about the focus of the introduction which need to be streamlined to focus on the aim of the study, the choice of thresholds used in the study and need for the authors to present the full error-bars of the climate impacts from the 17 GCMs. In addition, the authors should justify the choice of only 10 year old plantations, ensure that sampling bias correction of the presence points, relate the variables to the crop being modeled (have agronomic rather than biophysical variables) and also ensure that the maps correspond to the suitable/unsuitable partitioning that was presented in the method section. In addition, and most importantly, the authors present VIF scores as variable importance metrics and this should be corrected and the results on variable importance appropriately presented and discussed.

We look forward to receiving your revised manuscript.

Kind regards,

Abel Chemura

Academic Editor

PLOS ONE

Journal Requirements:

 “YES.  The research was funded by The Open University and The UK Research and Innovation through Global Challenges Research Fund (GCRF).”          

“The authors are grateful to the Open University and the UK Research and Innovation through Global Challenges Research Fund (GCRF) project for funding and academic guidance.”

 “YES.  The research was funded by The Open University and The UK Research and Innovation through Global Challenges Research Fund (GCRF).”

“NO”.

5. We note that Figure 1, 2, 3, 4 and 5 in your submission contain map images which may be copyrighted. All PLOS content is published under the Creative Commons Attribution License (CC BY 4.0), which means that the manuscript, images, and Supporting Information files will be freely available online, and any third party is permitted to access, download, copy, distribute, and use these materials in any way, even commercially, with proper attribution. For these reasons, we cannot publish previously copyrighted maps or satellite images created using proprietary data, such as Google software (Google Maps, Street View, and Earth). For more information, see our copyright guidelines: http://journals.plos.org/plosone/s/licenses-and-copyright.

a. You may seek permission from the original copyright holder of Figure 1, 2, 3, 4 and 5 to publish the content specifically under the CC BY 4.0 license. 

Additional Editor Comments (if provided):

scientific name of the crop can be added in the titleLine 17-19: the aim should capture both the current and future suitability modelling.Line 29: remove "the climate outputs", GCM always produce climate outputs and nothing elseLine 23-27: The results presented are conflicting, they need to be revised. The first sentence (Line 23) say that  "large parts of Malawi's macadamia growing regions will remain suitable for macadamia" but line 25 say that "suitable areas for macadamia production are predicted to shrink". The results need to be consistent and not confusing to the readers.Line 29: Most? That is a qualitative result, probably based on visual assessments. Quantitative results are more conclusive.line 55: Change will to may. its on the balance of probability.line 75-93: This information may not be necessary in the introduction. Be succinct and focused on the aim of the study and avoid broad issuesLine 99-100:: 98-100: Yes indeed, macadamia could have been omitted but that is not sufficient motivation for the study. Authors to clearly state the gap they are filling with their study (especially in light of the work by Evans, N. (2008). Suitability Mapping of the Malawi macadamia industry. *Irish Aid: Dublin, Ireland*). There is need for this study but it is not well articulated in the introduction. Line 118: Detailed and summary are conflicting words, it is not possible to have a detailed summary, a summary is a short version of the detailed report. it is also not a good idea to have reference to Supplementary materials in the Introduction section.Line 120: It is an important point that the perennial crops also store carbon and therefore an important aspect in climate mitigation. This important aspect is not well articulated in the introduction and just appears hanging here. You can further explain on this important aspect of the suitability study as part of your justification/motivation.Line 127: It will be good to indicate who and how the results can be used in climate planningLine 127: Overall introduction: The introduction is too long, winding and need to be more focused.Line 157: Since Malawi is not on the equator, what is the equivalent in Malawi?Line 162: RCP4.5 is not the optimistic scenario, the optimistic scenario is RCP2.6, which you did not use. it is important to be consistent so that we avoid confusing readersLine 165: 2050s (period 2046–2065): Kindly recheck if this is correct, it could be 2040-2060. Line 187: Justify the choice of the 0.77 threshold. Line 193: "Hence, absence data may not represent naturally occurring phenomena" This is not clear, revise.Line 196: 500 randomly generated pseudo-absence Line 221: The criteria to assess GCMs is to measure their performance on historical measurements and see which GCM is able to capture the trends and rhythms of the data. Line 227: Section 3.1: Authors should recheck the results as they appear to be from VIF and not variable importance. The results and related discussion should therefore be revised accordingly.Table 1: Move to Methods (and also add at the end of the manuscript and not inside text0Line 242: The plants will most likely not be able to grow on water, so the percentage of ONLY land area is more representative.Line 243-249: The descriptions of the colors should be below the Figure and not in the results. In addition these results are confusing because in the methods the authors say they used one threshold to get suitable and unsuitable areas but here they have 5 suitability classes. They should stick with their two classes whose methods are explained or explain in the methods section how the categories were generated. In addition, the authors present the results in terms of elevation but this is not one of the variables used in the modelling. All results using the 5 classes are therefore difficult to comprehend and understand if the choice of the classes are not scientifically justified.Line 247-251. The sentences are conflicting and should be revised. First authors claim that optimal suitability is in areas not exceeding 30 degrees (line 249) and then went on to mention that some optimal suitable areas are in areas above 30 degrees (line 251). This should be revised.Line 259: Remove "Though, these areas are also used for the production of other crops, particularly annuals". The results sections should be reserved to the results of the study and not anything else. Line 267-268: This is now discussion and should be moved to the discussion section.Line 270: " significant reductions" this is qualitative (and subjective), quantitative results are more convincing, what is the loss in suitability in the district, in numbers.Line 271-273: Leave that for the discussion section.Line 280: RCP4.5 is not the optimistic scenario.Line 283-284: Not clear, revise.Line 284-285: Explanations should be reserved for discussion section.Line 293-317: The authors should recheck the variable importance results and revise this section accordingly.Line 332-334: Revise the statement for clarity/senseLine 335 - 343: It will be important if the authors discuss their results with regard to how the impacts are likely to occur on the crop. for example which phenological stage is likely to be most affected and what is the effect of increased temperature on growth and production capacity of the plants, what happens when water is not enough or when water is not supplied during flowing. This makes the scientific basis of the study strong.

Reviewers' comments:

Reviewer's Responses to Questions

**Comments to the Author**

1. Is the manuscript technically sound, and do the data support the conclusions?

Reviewer #1: Yes

Reviewer #2: Yes

2. Has the statistical analysis been performed appropriately and rigorously? 

Reviewer #1: Yes

Reviewer #2: Yes

3. Have the authors made all data underlying the findings in their manuscript fully available?

Reviewer #1: Yes

Reviewer #2: Yes

4. Is the manuscript presented in an intelligible fashion and written in standard English?

Reviewer #1: Yes

Reviewer #2: Yes

5. Review Comments to the Author

Reviewer #1: Is the manuscript technically sound?

Yes, the subject of climate change and its impact on agricultural systems cannot be ignored any longer especially in SSA, where livelihoods of about two third of smallholder farmers are vulnerable to negative effects of global warming on water availability. The modelling framework and the synthesis of results makes the information contained in the manuscript useful in strategic planning of adaptation approaches for macadamia production in Malawi. The information is a valuable decision support tool for policy makers.

Do the data support the conclusions?

I am little skeptical about the the criteria used for selection of districts and representativeness of the 120 orchards sampled. The close proximity of orchards in some districts (Fig 3) is troublesome for me. Furthermore, authors sampled 10-year old macadamia orchards yet they do not provide the justification for this decision. As it looks, exclusion of <10 and >10 year orchards introduces bias in the dataset. My argument is that an orchard is an orchard irrespective of how old. I now wonder if using pseudo-absence points was to conceal this sampling limitation. On the other hand, since the outputs highlight trends that authors have ably discussed with research literature from Malawi makes me think the data supports the conclusions.

Has the statistical analysis been performed appropriately and rigorously?

Yes. The authors seem to have good understanding of the analytical steps performed. I applaud them for informed technical judgement made before using the presence-only approach of analysis (line 190b-192a). However, they sometimes forget to provide support to why somethings were done. There is a confusion of terminology e.g., performance and accuracy. In practice, accuracy should be considered as the measure of performance.

Have the authors made all data underlying the findings in their manuscript fully available?

Some information underlying the findings in the manuscript are available on https://zenodo.org/record/4751439

Is the manuscript presented in an intelligible fashion and written in standard English?

Yes, the manuscript is presented in standard English. However, some paragraphs of the introduction need shortening as several parts could benefit the site description in materials and methods. Authors should adopt an inverted equilateral triangle structure.

Reviewer #2: Authors used an ensemble model to determine the current and future distribution of Macadamia producing areas in Malawi. They used 17 GCMs based on two scenarios i.e., RCP4.5 and RCP8.5. Such studies as presented by these authors are critical particularly in Africa where most of the economies are Agrarian based and are likely to be affected by climate change. The paper is well written and properly justified with the methodologies showing no flaws according to scientific standards.

1. However, the authors need to structure the flow of ideas, concepts and motivation in the introduction section where few redundant ideas were observed.

2. The authors must justify why they used the RCP 4.5 and 8.5 and not any of the other scenarios

3. Provide a statistical justification of the use of 500 background pseudo-absence points and not the widely used 10 000 points (line 196)

4. Authors must expand and clarify on the 17 GCMs used enough to be repeatable. Which ones were used and why?

5. Revise a few inconsistences observed regarding the capitalization of acronyms.

6. PLOS authors have the option to publish the peer review history of their article (what does this mean?). If published, this will include your full peer review and any attached files.

Reviewer #1: No

Reviewer #2: No

---

## [Author Response · Author response to Decision Letter 0]

30 Jul 2021

1. scientific name of the crop can be added in the title

 - Macadamia integrifolia added.

2.Line 17-19: the aim should capture both the current and future suitability modelling.

 - This has bee revised accodingly.

3. Line 23-27: The results presented are conflicting, they need to be revised. The first sentence (Line 23) say that "large parts of Malawi's macadamia growing regions will remain suitable for macadamia" but line 25 say that "suitable areas for macadamia production are predicted to shrink". The results need to be consistent and not confusing to the readers.

 - The results have been revised and have made sure that there is no confliction.

4. Line 29: Most? That is a qualitative result, probably based on visual assessments. Quantitative results are more conclusive.

 - This has been rectified and we made sure that there are percentages and figures in the revision.

5. line 55: Change will to may. its on the balance of probability.

 - Done.

6. line 75-93: This information may not be necessary in the introduction. Be succinct and focused on the aim of the study and avoid broad issues.

 - The introduction has been re-writen to make sure its foccussed to the research question.

7. Line 99-100:: 98-100: Yes indeed, macadamia could have been omitted but that is not sufficient motivation for the study. Authors to clearly state the gap they are filling with their study (especially in light of the work by Evans, N. (2008). Suitability Mapping of the Malawi macadamia industry. Irish Aid: Dublin, Ireland). There is need for this study but it is not well articulated in the introduction. 

 - This has been addresed.

8. Line 118: Detailed and summary are conflicting words, it is not possible to have a detailed summary, a summary is a short version of the detailed report. it is also not a good idea to have reference to Supplementary materials in the Introduction section.

 - Edited and corrected.

9. Line 120: It is an important point that the perennial crops also store carbon and therefore an important aspect in climate mitigation. This important aspect is not well articulated in the introduction and just appears hanging here. You can further explain on this important aspect of the suitability study as part of your justification/motivation.

 - This has been taken to consideration and have included it in the justification.

10. Line 127: It will be good to indicate who and how the results can be used in climate planning.

 - We have included a sentence that shows this.

11. Line 127: Overall introduction: The introduction is too long, winding and need to be more focused.

 - This has been shortened.

12. Line 157: Since Malawi is not on the equator, what is the equivalent in Malawi?

- Edited to reflect changes.

13. Line 162: RCP4.5 is not the optimistic scenario, the optimistic scenario is RCP2.6, which you did not use. it is important to be consistent so that we avoid confusing readers.

 - Corrected.

14 Line 165: 2050s (period 2046–2065): Kindly recheck if this is correct, it could be 2040-2060.

 - Corrected.

15. Line 187: Justify the choice of the 0.77 threshold.

 - We have justified why we used this threshold based on previous studies.

16. Line 193: "Hence, absence data may not represent naturally occurring phenomena" This is not clear, revise.

 - Has been revised.

17. Line 196: 500 randomly generated pseudo-absence

 - Addressed in the cover letter.

18. Line 221: The criteria to assess GCMs is to measure their performance on historical measurements and see which GCM is able to capture the trends and rhythms of the data.

 - Noted.

19. Line 227: Section 3.1: Authors should recheck the results as they appear to be from VIF and not variable importance. The results and related discussion should therefore be revised accordingly.

 - We checked this and have included the results on this as a graph.

20. Table 1: Move to Methods (and also add at the end of the manuscript and not inside text0

- This has been done according to the advice.

21. Line 242: The plants will most likely not be able to grow on water, so the percentage of ONLY land area is more representative.

 - The number presented is for surface of the land excluding the waters.

22. Line 243-249: The descriptions of the colors should be below the Figure and not in the results. In addition these results are confusing because in the methods the authors say they used one threshold to get suitable and unsuitable areas but here they have 5 suitability classes. They should stick with their two classes whose methods are explained or explain in the methods section how the categories were generated. In addition, the authors present the results in terms of elevation but this is not one of the variables used in the modelling. All results using the 5 classes are therefore difficult to comprehend and understand if the choice of the classes are not scientifically justified.

 - This has been addressed.

22. Line 247-251. The sentences are conflicting and should be revised. First authors claim that optimal suitability is in areas not exceeding 30 degrees (line 249) and then went on to mention that some optimal suitable areas are in areas above 30 degrees (line 251). This should be revised.

 - This has been revised.

23. Line 259: Remove "Though, these areas are also used for the production of other crops, particularly annuals". The results sections should be reserved to the results of the study and not anything else.

 - Done.

24. Line 267-268: This is now discussion and should be moved to the discussion section.

- Removed to discussion.

25. Line 270: " significant reductions" this is qualitative (and subjective), quantitative results are more convincing, what is the loss in suitability in the district, in numbers.

- Addressed in the revised manuscript.

26. Line 271-273: Leave that for the discussion section.

- Moved to discussion.

27. Line 280: RCP4.5 is not the optimistic scenario.

- Revised.

28. Line 283-284: Not clear, revise.

 - Corrected.

29. Line 284-285: Explanations should be reserved for discussion section.

- Addressed.

30. Line 293-317: The authors should recheck the variable importance results and revise this section accordingly.

 - Revised.

31. Line 332-334: Revise the statement for clarity/sense

 - Clarified.

32. Line 335 - 343: It will be important if the authors discuss their results with regard to how the impacts are likely to occur on the crop. for example which phenological stage is likely to be most affected and what is the effect of increased temperature on growth and production capacity of the plants, what happens when water is not enough or when water is not supplied during flowing. This makes the scientific basis of the study strong.

 - This has been done.

---

## [Decision Letter · Decision Letter 1]

23 Aug 2021

Climate suitability predictions for the cultivation of macadamia (Macadamia integrifolia) in Malawi using climate change scenarios.

PONE-D-21-15846R1

Dear Dr. Zuza,

We’re pleased to inform you that your manuscript has been judged scientifically suitable for publication and will be formally accepted for publication once it meets all outstanding technical requirements.

Kind regards,

Abel Chemura

Academic Editor

PLOS ONE

Additional Editor Comments (optional):

Reviewers' comments:

Reviewer's Responses to Questions

**Comments to the Author**

1. If the authors have adequately addressed your comments raised in a previous round of review and you feel that this manuscript is now acceptable for publication, you may indicate that here to bypass the “Comments to the Author” section, enter your conflict of interest statement in the “Confidential to Editor” section, and submit your "Accept" recommendation.

Reviewer #1: All comments have been addressed

Reviewer #2: (No Response)

2. Is the manuscript technically sound, and do the data support the conclusions?

Reviewer #1: Yes

Reviewer #2: (No Response)

3. Has the statistical analysis been performed appropriately and rigorously? 

Reviewer #1: Yes

Reviewer #2: (No Response)

4. Have the authors made all data underlying the findings in their manuscript fully available?

Reviewer #1: Yes

Reviewer #2: (No Response)

5. Is the manuscript presented in an intelligible fashion and written in standard English?

Reviewer #1: Yes

Reviewer #2: (No Response)

6. Review Comments to the Author

Reviewer #1: The authors have satisfactorily responded to the reviewer comments and addressed the concerns raised. The revised manuscript is refined with most of the earlier ambiguity removed from the different sections. I believe the manuscript is technically sound and statistical analyses performed are standard for geospatial modelling and machine learning.

Introduction

- Long sentence in line 40b - 43 should be split into two. For example, "Malawi is particularly vulnerable to climate change because of its high poverty level, limited cash flow and technological infrastructure. Moreover, the country is heavily reliant on the rain-fed agricultural sector, which is the backbone of the economy and society". This example accounts for the first part of line 44 (can be deleted).

- Line 64 not sure citing Tables in the introduction is among the acceptable practices for PLoS journals. The reference [21] will suffice.

- Line 78-90 I believe comprises the aim, objectives and justification for the study. But, why not simply make it explicit like is traditional practice. Keeping it simple and easy for the reader to know what the study was all about.

As a side note: If not already used, I recommend the authors to use the referencing system (e.g. Endnote or Mendeley) to enable seamless citation in accordance with PLoS referencing style.

Methodology

- TSS (True Skills Statistic) needs to be written in full when mentioned for the first time.

Results

– Are consistent and coherent with the objectives and methodology used.

Discussion

- Ok

Reviewer #2: (No Response)

7. PLOS authors have the option to publish the peer review history of their article (what does this mean?). If published, this will include your full peer review and any attached files.

Reviewer #1: No

Reviewer #2: No

---

## [Editor Report · Acceptance letter]

31 Aug 2021

PONE-D-21-15846R1 

Climate suitability predictions for the cultivation of macadamia *(Macadamia integrifolia)* in Malawi using climate change scenarios. 

Dear Dr. Zuza:

I'm pleased to inform you that your manuscript has been deemed suitable for publication in PLOS ONE. Congratulations! Your manuscript is now with our production department. 

Kind regards, 

on behalf of

Dr. Abel Chemura 

Academic Editor

PLOS ONE